# Long-Term Evaluation of Retinal Morphology and Function in Rosa26-Cas9 Knock-In Mice

**DOI:** 10.3390/ijms24065186

**Published:** 2023-03-08

**Authors:** Kabhilan Mohan, Sushil Kumar Dubey, Kyungsik Jung, Rashmi Dubey, Qing Jun Wang, Subhash Prajapati, Jacob Roney, Jennifer Abney, Mark Ellsworth Kleinman

**Affiliations:** 1Department of Surgery, East Tennessee State University, Johnson City, TN 37614, USAdubeys@etsu.edu (S.K.D.);; 2Department of Ophthalmology and Visual Sciences, University of Kentucky, Lexington, KY 40536, USA; 3Department of Biochemistry and Molecular Genetics, University of Virginia, Charlottesville, VA 22908, USA

**Keywords:** SpCas9, genome editing, retina, retinal pigment epithelium, long-term safety, Cas9 knock-in mice

## Abstract

The CRISPR/Cas9 system is a robust, efficient, and cost-effective gene editing tool widely adopted in translational studies of ocular diseases. However, in vivo CRISPR-based editing in animal models poses challenges such as the efficient delivery of the CRISPR components in viral vectors with limited packaging capacity and a Cas9-associated immune response. Using a germline Cas9-expressing mouse model would help to overcome these limitations. Here, we evaluated the long-term effects of SpCas9 expression on retinal morphology and function using Rosa26-Cas9 knock-in mice. We observed abundant SpCas9 expression in the RPE and retina of Rosa26-Cas9 mice using the real-time polymerase chain reaction (RT-PCR), Western blotting, and immunostaining. SD-OCT imaging and histological analysis of the RPE, retinal layers, and vasculature showed no apparent structural abnormalities in adult and aged Cas9 mice. Full-field electroretinogram of adult and aged Cas9 mice showed no long-term functional changes in the retinal tissues because of constitutive Cas9 expression. The current study showed that both the retina and RPE maintain their phenotypic and functional features in Cas9 knock-in mice, establishing this as an ideal animal model for developing therapeutics for retinal diseases.

## 1. Introduction

In the past few years, clustered, regularly interspaced, short palindromic repeats (CRISPR)-based genome editing has found widespread research and clinical applications because of its specificity, affordability, multiplexing, and ease of use [1,2,3,4,5]. The type II CRISPR system, employing the Streptococcus pyogenes-derived Cas9 endonuclease (SpCas9), is the best-characterized CRISPR system to date [6,7]. The Cas9 protein, complexes with a single guide RNA (sgRNA), a synthetic fusion of naturally occurring CRISPR RNA (crRNA) and transactivating crRNA, performs the sequence-specific targeting and cleavage of DNA to generate precise double-stranded breaks (DSBs) [2,3,8]. These DSBs drive genome editing via one of the DNA repair pathways, namely non-homologous end joining or homology-directed repair [9,10,11,12]. The potential to direct Cas9 to virtually any target of interest through the design of synthetic sgRNAs has accelerated its use for genome editing.

Several studies have successfully applied the CRISPR/Cas9 tool to dissect the genetic underpinnings of ocular disease mechanisms and detected potential therapeutic targets [13,14,15,16]. Despite the vast applications of CRISPR technology for treating various genetic and multifactorial ocular diseases, there are still concerns about Cas9-associated toxicity related to tissue integrity, visual function, and immunogenicity. It is unclear whether the expression of the bacterially derived Cas9 protein is tolerated in different retinal cell types. There have been a few studies that have evaluated the effects of Cas9 expression in ocular tissues [17,18,19,20]. SpCas9 expression in the retina after intravitreal injections of adeno-associated virus (AAV) vectors carrying the CRISPR/Cas9 machinery was studied. No retinotoxicity was observed in the mice for five weeks post-injection; however, the possibility of there being detrimental effects on the retina due to long-term SpCas9 expression remains unexplored [18]. Subsequently, Jo et al. delivered intravitreal injections of AAV vectors encoding an SpCas9 ortholog, CjCas9 (Campylobacter jejuni-derived Cas9), and observed no abnormality in retinal histology and function up to 14 months post-injection [19]. Some preclinical studies in various tissues, including the retina, have shown that despite the success achieved with CRISPR-mediated gene editing, there was sporadic Cas9 associated immune activation without extensive cell death [17,20]. Furthermore, the efficacy of in vivo applications of CRISPR is also limited by various factors such as packaging the bulky SpCas9, along with sgRNA inside a single viral delivery vector such as AAV, and the technical variability in independent injection practices. This prompted several studies to explore various SpCas9 orthologs as alternative CRISPR tools, and a few researchers have demonstrated successful genome editing in retinal cells [16,19,21,22]. However, Li et al. recently showed that SpCas9 had the highest efficacy compared to those of other Cas9 endonucleases, both in vivo and in vitro [23].

The challenge of delivering SpCas9 to different retinal cell types can be eliminated using Cas9 knock-in animal models that express SpCas9 constitutively and can prove to be a viable option for gene editing [24]. Additionally, germline Cas9-expressing mice alleviate any concerns of immunogenic response of the host tissue towards the prokaryotic Cas9 protein. In this context, our study evaluated the long-term effects of SpCas9 expression on the RPE and retinal morphology and function in adult (4–6 months) and aged (10–14 months) Rosa26-Cas9 knock-in mice (herein referred to simply as Cas9 mice). The Cas9 mouse model utilized in this study was developed in the C57BL/6J background by introducing a Cas9 expression cassette into the Rosa26 locus. This cassette includes a 3X FLAG-tagged SpCas9 linked to an EGFP gene under the control of a ubiquitous CAG promoter [24]. The Cas9knock-in mouse can facilitate efficient genome editing using viral or non-viral sgRNA delivery methods, providing an attractive model for CRISPR applications in genetic and multifactorial ocular disorders.

## 2. Results

### 2.1. Cas9 Knock-In Mice Exhibit Ubiquitous Expression of SpCas9 in RPE and Retinal Layers

Cas9 mice constitutively express Cas9-EGFP under the control of a ubiquitously expressed CAG promoter. A sagittal ocular section (anterior to posterior) of a Cas9 mouse showed Cas9-EGFP expression around the entire circumference of the eye (Appendix A). qPCR data from Cas9 and wild-type (WT, C57BL/6J) mice showed a substantial expression of SpCas9 mRNA in the RPE/choroid and neural retina (Figure 1A; Cas9 mice n = 6, WT mice n = 5) only in the Cas9 mice. These results were further corroborated by Western blot analyses, which demonstrated high levels of SpCas9 protein in both the RPE/choroid and neural retina (Figure 1B; n = 3 per group) of Cas9 mice. Subsequently, we compared the expression levels of Cas9 in adult and aged Cas9 mice with the housekeeping gene, beta-actin, as well as RPE-specific gene, *Rpe65*, and retina-specific genes, *Rho* and *Prph2*. Our findings indicate that in both adult and aged RPE, Cas9 expression is comparable to beta-actin, but the level of it is significantly higher than that of the *Rpe65* gene (Figure 1C). Likewise, in the retina, the expression of Cas9 is similar to that of the actin gene, but it is considerably lower than the highly expressed *Rho* and *Prph2* genes. Furthermore, we observed that the expression of Cas9 is sustained in aged mice relative to that of beta-actin in both types of tissues (Figure 1C).

Fluorescence stereomicroscope images of retinal flat mounts obtained from constitutive Cas9-expressing and WT mice showed Cas9-EGFP expression throughout the retina in adult and aged Cas9 mice (Figure 1D). Cas9-EGFP expression was visualized in more detail by the fluorescence imaging of frozen retinal sections from WT (n = 3 per group) and Cas9 mice (n = 3 per group). EGFP expression was ubiquitously seen across the different retinal layers (Figure 1E). Additionally, we performed immunofluorescence (IF) staining of frozen eye sections obtained from adult and aged WT (n = 3 per group) and Cas9 mice (n = 3 per group) to directly visualize SpCas9 expression in the retina. Cas9 mice stained positive for SpCas9 protein with a global expression in different layers of the retina. As expected, WT mice stained negative for SpCas9 (Figure 1F). IF and EGFP images revealed that the Cas9 expression across the retinal layers was not uniform and appears to be higher in the ganglion cell layer (GCL), inner nuclear layer (INL), and inner segment of the photoreceptor layer (PRL). Together, these findings confirmed global SpCas9 expression in Cas9 knock-in mice at both the mRNA and protein levels in the RPE and neural retina. The expression of Cas9 protein remained stable in Cas9 mice throughout their adult and aged stages.

### 2.2. Effects of Long-Term SpCas9 Expression on Retinal Phenotype in Adult and Aged Cas9 Mice

We characterized the retinal phenotypes in adult (4–6 months) and aged (10–14 months) Cas9 mice and compared them with those of the age-matched WT controls. These mice also tested negative for spontaneous rd8 mutation. The adult Cas9 mice exhibit normal fundi with regular retinal vasculature (n = 6–8 per group, Figure 2A, WT; Figure 2E, Cas9 mice). Consistently, the fundus and retinal autofluorescence of the aged Cas9 mice (n = 6) were comparable to those of the age-matched WT mice (Figure 2B,D, WT; Figure 2F,H, Cas9 mice). Next, we performed fluorescein angiography to visualize the architecture of retinal blood vessels. Angiography revealed normal vasculature in both the WT and Cas9 mice (Figure 2C, WT; Figure 2G, Cas9 mice) with no damaged or leaking blood vessels. Antibodies directed against the tight junction protein zona occludens-1 (ZO-1) showed regular hexagonal tiling of the RPE in the adult WT and Cas9 cohorts (n = 4; Figure 2I, WT; Figure 2K, Cas9 mice). The RPE morphology of aged Cas9-expressing mice appeared to be similar to that of the control groups (n = 3; Figure 2J, WT; Figure 2L, Cas9 mice). We observed nuclear localized EGFP signal corresponding to Cas9 in both adult and aged Cas9 mice. However, the EGFP expression appeared to be a mosaic, probably owing to decreased imaging sensitivity at a lower magnification (Figure 2K,L). We observed EGFP positivity in nearly all the cells at a higher magnification (Figure 2K,L—inset). Collectively, these data suggest that RPE and retinal phenotype appeared to be well preserved in adult and aged mice expressing SpCas9 protein.

### 2.3. Examination of Retinal Thickness and Histologic Integrity in Mice with Endogenous Expression of SpCas9

The Cas9 mice were assessed for spatiotemporal changes in retinal morphology and thickness using spectral domain optical coherence tomography (SD-OCT) imaging. Images from both the WT and Cas9 mice revealed clearly defined retinal layers with no apparent structural abnormalities (Figure 3A, WT adult; Figure 3B, Cas9 adult; Figure 3C, Cas9 aged). Volume scans centered on the optic nerve were obtained, and the average retinal thickness within a 3 mm radius was measured. Multiple measurements were automatically averaged to obtain thickness at radial positions for each eye (Appendix A). These data were then averaged across eyes to obtain the final retinal thickness measurement. No significant difference in total retinal thickness was observed between the WT (243.1 ± 6.06 µm, mean ± SD, n = 8 eyes, adult) and Cas9 (250.1 ± 11.23 µm, n = 6 eyes, adult; 245.1 ± 5.74 µm, n = 8 eyes, aged) mice (Figure 3D; *p* = 0.53, Kruskal–Wallis test). The H&E-stained sections did not reveal significant abnormalities in the Cas9 mouse retina, which appeared to be phenotypically similar to those of the wild-type retina (Figure 3E, WT adult; Figure 3F, Cas9 adult; Figure 3G, Cas9 aged).

### 2.4. Assessment of Retinal Function by Electroretinography

To evaluate the potential toxicity of pervasive SpCas9 expression to retinal function, we compared the full-field electroretinogram (ERG) responses to different flash intensities in adult and aged Cas9 mice with those of the age-matched WT controls. Scotopic ERG waveforms appeared to be largely normal in the Cas9 mice (n = 8 eyes, adult; n = 6 eyes, aged) compared to those of the WT mice (n = 6 eyes, adult; n = 6 eyes, aged; Figure 4A). ERG recordings from the eyes of Cas9 mice and age-matched controls showed no significant difference in scotopic a- and b-wave responses across a wide range of flash intensities, indicating no loss of retinal function (Figure 4B,C). The retinal function remained quite stable even in the aged mice, and the ERG results support that long-term Cas9 expression does not compromise retinal response to various ERG amplitudes. As summarized in Table 1 and Table 2, the statistical analysis using Mann–Whitney U test indicates that there were no significant changes in the ERG amplitudes and implicit times between WT and Cas9 mice in both the adult and aged groups.

## 3. Discussion

CRISPR-Cas9 is a powerful tool used for sequence-specific genome engineering, and it is being rapidly adopted in ophthalmology, including gene therapy for Leber’s congenital amaurosis, glaucoma, age-related macular degeneration, retinitis pigmentosa, Usher syndrome, achromatopsia, and other eye diseases [14,15,16,25,26,27,28,29]. The eye exhibits a degree of immune privilege and tight blood–ocular barriers, making it the ideal organ for clinical translation studies (NCT03872479, NCT03748784, and NCT03066258) [30]. Despite several ongoing retinal gene therapy studies, there have been recent reports of host cellular and humoral immune response to Cas9 in animal models, which could pose a critical barrier for CRISPR-based applications [31,32,33]. Animal models expressing Cas9 from early embryonic stages and throughout their life-span can circumvent any host immune response to Cas9. A previous study in Rosa26-Cas9 knock-in mouse showed that constitutive SpCas9 expression did not result in any detectable toxicity or morphological abnormalities in multiple organs (including the brain, kidney, liver, spleen, and heart tissue) of this transgenic strain [24]. Multiple studies have provided evidence of the safety of Cas9 mice by showcasing the utilization of diverse tissues and cells from these murine models [34,35,36]. However, long-term SpCas9 expression and its associated toxicity have not been characterized in ocular tissues in these mice, which can be bred for germline transmission of the Cas9 allele.

The retinal tissues are highly sensitive to transgene expression levels, and the strong expression of genes under a heterologous promoter can cause toxicity to the retinal cells [37,38,39,40,41]. Additionally, the SpCas9 protein has no specific function assigned in the RPE and retinal tissues and is not a part of the normal retinal proteome. Additionally, the processing and turnover mechanism of the abundantly expressed SpCas9 in the retinal cells is unclear. There is a possibility of SpCas9 protein being poorly degraded or accumulating in the cells because of the inefficient turnover mechanism. This could alter the function of other endogenous proteins, leading to dysregulation of the cellular processes, cytotoxicity, and inflammation. Therefore, with our Cas9 mouse model showing a strong and persistent Cas9 expression, it becomes critical to evaluate any changes in the retinal structure, integrity, and function induced by the long-term expression of SpCas9.

The goal of the current study is to address the long-term safety concern of SpCas9 expression in the neural retina and RPE/choroid in Cas9 knock-in mouse model. The Cas9 mice express Cas9 constitutively under the control of the CAG promoter, in addition to EGFP and a 3X FLAG epitope tag. This enables the detection of Cas9 expression indirectly via EGFP fluorescence or anti-FLAG antibodies. EGFP fluorescence was observed on the retinal (Figure 1D) and RPE/choroid (Figure 2K,L) flat mounts, as well as around the entire circumference of the eye (Appendix A). We also confirmed abundant SpCas9 expression at the transcriptional and protein levels in the RPE/choroid and neural retina of Cas9 knock-in mice (Figure 1A,B,F). The interrogation of the retinal structure and vasculature using SD-OCT and fundus imaging appeared to be normal in Cas9 mice. The ERG response and histology of the retina of Cas9 mice appeared to be normal. A limitation of this study is that ultra-structural studies of the photoreceptor and RPE layer by transmission electron microscopy in Cas9 mice were not performed.

The Cas9knock-in mouse model expresses Cas9 protein endogenously, which overcomes the challenge of immunogenic response in the host occasionally seen with exogenous Cas9 delivery systems [42,43]. The Cas9 knock-in mice are also particularly suited to address the difficulties that arise from in vivo delivery of the bulky SpCas9 molecule, the short half-life of Cas9 protein inside the cell, and optimizing the delivery methods [44,45,46]. Thus, this germline Cas9-expressing mice presents an attractive model for gene editing studies in the posterior segment of the eye.

In conclusion, our data suggest that the Rosa26-Cas9 knock-in mouse is an excellent model system for genetic manipulation in the RPE/choroid and neural retina using CRISPR tools. To the best of our knowledge, no study has thus far comprehensively assessed toxicity due to prolonged SpCas9 expression in the posterior segment of the eye. Our data demonstrate that sustained SpCas9 expression appears to be well tolerated in the RPE/choroid and neural retinal with no loss in tissue integrity and function. These findings have important implications in determining the safety of a Cas9 knock-in mouse and set the stage for potential future Cas9-mediated retinal gene therapies.

## 4. Materials and Methods

### 4.1. Mice

All the animal experiments were approved by the University of Kentucky and East Tennessee State University Institutional Animal Care and Use Committee and were conducted in accordance with the Association for Research in Vision and Ophthalmology Statement for the Use of Animals in Ophthalmic and Visual Research. Male and female wild-type C57BL/6J mice (Stock No: 000664; Jackson Laboratory, Bar Harbor, ME, USA) and Rosa26-Cas9 knock-in mice (Stock No: 026179; Jackson Laboratory, Bar Harbor, ME, USA) on the C57BL/6J background between 4 and 14 months of age were used for the experiments.

### 4.2. Rd8 Sequencing in Cas9 Mice

The method for Rd8 sequencing in Cas9 mice was described in the study by Wang et al. [47]. In brief, genomic DNAs were prepared from ear punches by isopropanol precipitation. PCR reactions were carried out using the Rd8 primers (Forward: 5′-GGT GAC CAA TCT GTT GAC AAT CC-3′; Reverse: 5′-GCC CCA TTT GCA CAC TGA TGA C-3′; annealing temperature 55 °C). The PCR amplicons (~434 bp) were resolved on a 1% agarose gel, extracted and sequenced for the Rd8 mutation (i.e., the single-base deletion mutation c.3481delC in the Crb1 gene) by Sanger sequencing.

### 4.3. Mice Anesthesia

Mice were anesthetized using intraperitoneal injections of a ketamine-xylazine cocktail (ketamine hydrochloride 100 mg/kg, xylazine 10 mg/kg; Henry Schein Animal Health, Dublin, OH, USA). Pupils were dilated using 1% tropicamide (Akorn, IL, USA) and 2.5% phenylephrine chloride (Akorn, IL, USA). Proparacaine (Akorn, IL, USA) was used as a topical anesthetic to avoid corneal reflex during injections. Following the injections, a polymyxin/neomycin b sulfate bacitracin zinc ophthalmic ointment (Akorn, IL, USA) was applied at the surgical site to minimize the risk of infections.

### 4.4. Fundus Imaging

Following pupillary dilation, fundus photographs were taken of unanesthetized mice using appropriate filters using the TRC 50-IX camera (Topcon, Japan) integrated with the OphthaVision digital imaging suite (Sonomed Escalon). Images were exported from the Topcon imaging suite as TIFF files and prepared using Adobe Photoshop (Adobe, San Jose, CA, USA).

### 4.5. Electroretinography

Mice were dark adapted overnight to assess the scotopic full-field ERG response. Experiments were conducted with the aid of dim red lighting. Following anesthesia, the mice were placed on a pre-warmed heating table, and both eyes were in contact with light guide electrodes that function simultaneously as a stimulator and an electrode (Diagnosys Celeris; Diagnosys LLC, Lowell, MA, USA). A reference skin electrode was placed at the base of the head and a ground electrode at the base of the tail. Electrodes were impedance matched between both eyes. The eyes were kept moist using methylcellulose solution (Goniovisc; Sigma Pharmaceuticals, IA, USA). Mouse eyes were then exposed to increasing intensity flashes of light, triggered using the Espion software (Espion v3) with a preset automatic protocol. Multiple flashes were averaged to obtain a single waveform at each recorded intensity.

### 4.6. Spectral-Domain Optical Coherence Tomography (SD-OCT)

Mice were anesthetized and placed on a pre-warmed heating pad to maintain a constant body temperature. Their eyes were dilated, and corneal hydration was maintained by the liberal application of Systane Ultra (Alcon, Fort Worth, TX, USA). SD-OCT was performed using the Spectralis HRA-OCT imaging platform (Heidelberg Engineering, Heidelberg, Germany), and volume scans were obtained centered on the optic nerve. Images were recorded using the automatic real-time mode, and at least 15 consecutive images were averaged to obtain a single scan. The images were manually segmented and retinal thickness information was obtained 1 mm, 2 mm, and 3 mm from the optic nerve. Images were also exported as TIFF files for representation.

### 4.7. Flat Mount Preparation and Immunofluorescence

Eyes were enucleated post-euthanization and fixed in 2% paraformaldehyde (PFA) for 30 min. The anterior parts including the cornea, the lens, and the iris were removed. To avoid curling, radial cuts were made in a symmetrical four-leaf pattern. The resulting posterior eyecups were fixed in 2% PFA for 30 min at room temperature either for retinal or RPE flat mounts. For ZO-1 staining, flat mounts were fixed in 2% paraformaldehyde for 1 h, followed by PBS washes and blocking for 2 h in 3% bovine serum albumin and 4% goat serum at 4 °C. The flat mounts were then incubated with anti-ZO-1 (Invitrogen 61-7300, Rockford, IL, USA) overnight followed by washing, and fluorescent secondary antibody (F(ab’)2-goat anti-rabbit IgG (H + L) was cross-adsorbed, and the Alexa Fluor 594 (Invitrogen A-11072, Eugene, OR, USA) treatment was applied for 2 h at 4 °C. The flat mounts were mounted on slides using hard set mounting media (Vectashield H-1400, Vector Laboratories, Burlingame, CA, USA) and imaged using an Acousto-Optical Beam Splitter SP5 confocal microscope (AOBS SP5, Leica GmbH, Wetzlar, Germany).

Mouse frozen eye sections were fixed with 4% paraformaldehyde for 15 min, and then permeabilized in PBS with 0.25% Triton X-100 for 20 min at room temperature. Frozen sections were then blocked in PBS with 4% normal goat serum and 3% bovine serum albumin for 1 h, followed by incubation with M.O.M. Blocking Reagent (Vector Laboratories MKB-2213-1, Burlingame, CA, USA) for 1 h. These sections were incubated with anti-Cas9 antibody (1 μg/mL; Active Motif) conjugated with Alexa Fluor 594 (1:1000; Thermo Fisher Scientific A-11020, IL, USA) for 1 h, followed by nuclear stain with Hoechst 33342 (1:10,000; Invitrogen H3570, Eugene, OR, USA) for 5 min. Images were captured using the TCS SP8 confocal microscopy (Leica).

### 4.8. H&E Staining

Retinal sections (thickness, 5 µm) were prepared from paraffin-embedded eyes (n = 3 per group). After dewaxing and rehydration, the sections were stained with hematoxylin (VWR, 95057-858, Radnor, PA, USA) for 5 min, bluing (VWR, 95057-852) for 30 sec, and eosin for 30 sec (VWR, 95057-848). Stained slides were mounted by Cytoseal XYL (Thermo Scientific, 8312-4, Kalamazoo, MI, USA), and dehydrated. Representative images of the stained sections were obtained using a light microscope (Olympus, MA, USA).

### 4.9. qPCR

RNA extraction from neural retina and RPE and on-column genomic DNA removal was achieved using the Pure Link RNA Micro Kit (Invitrogen, Carlsbad, CA, USA) as per the manufacturer’s instructions. RNA was reverse-transcribed using the High Capacity cDNA Reverse Transcription Kit (Applied Biosystems 4368814, Lithuania), and then the resultant cDNAs were amplified using the Power SYBR Green PCR Master Mix (Applied Biosystems 4368706, Lithuania) and StepOnePlus Real-Time PCR System (Applied Biosystems, Lithuania). The expression of *Cas9*, *Rpe65*, *Rho* and *Prph2* genes was assessed using the primer sequences (Cas9FP:5′ AAACAGCAGATTCGCCTGGA 3′ and Cas9RP:5′ TCATCCGCTCGATGAAGCTC 3′; mRpe65FP:5′ AAGGCTCCTCAGCCTGAAGTCA 3′ and mRpe65RP:5′ GAGAACCTCAGGTTCCAGCCAT 3′; mRhoFP:5′ GAGGGCTTCTTT-GCCACACTTG 3′ and mRhoRP:5′ AGCGGAAGTTGCTCATCGGCTT 3′; mPrph2FP:5′ GCAATCGCTACCTGGACTTCTC 3′ and mPrph2RP:5′ GTGAGCTGGTACTGGATA-CAGG 3′) specific for these genes. Relative gene expression changes were calculated using either GAPDH (FP:5′ CATCACTGCCACCCAGAAGACTG 3′ and RP:5′ ATGCCAG-TGAGCTTCCCGTTCAG 3′) or beta-actin (FP:5′ GTGACGTTGACATCCGTAAAGA 3′ and RP:5′ GCCGGACTCATCGTACTCC 3′) for the mice as the reference gene. The fold change was calculated by determining the ratio of mRNA levels to control values using the Δ threshold cycle (Ct) method (2−ΔΔCt).

### 4.10. Western Blot

RPE/choroids and neural retinas from WT and Cas9 mice were lysed in 1x RIPA buffer (Thermo Scientific 89900, Rockford, IL, USA) containing protease (Thermo Scientific, A32963), and phosphatase (Thermo Scientific A32957, Rockford, IL, USA) inhibitor cocktails. Tissue lysates were quantified using the Rapid Gold BCA Protein Assay Kit (Pierce A53226, Rockford, IL, USA), and 25 µg protein was loaded onto 4–20% Tris-Glycine gels (Thermo Scientific XP04200BOX, Rockford, IL, USA). Proteins were transferred onto PVDF membrane (Millipore ISEQ00005, Burlington, MA, USA), blocked with 5% skimmed milk in phosphate-buffered saline (PBS) for 1 h, and incubated overnight at 4 °C in an orbital shaker using the following primary antibodies: anti-Cas9 (1:1000; Cell Signaling Technology 19526, Danvers, MA, USA) or anti-β-actin (1:1000; Cell Signaling Technology 4970, Danvers, MA, USA). Membranes were washed thrice with PBS-T and stained with HRP-conjugated secondary antibodies (1:5000; Invitrogen G21234, Carlsbad, CA, USA) for 2 h at room temperature. Stained membranes were washed thrice in PBS-T, and then developed using SuperSignal West Pico PLUS Chemiluminescent Substrate (Thermo Scientific 34580, Rockford, IL, USA) and imaged using a gel imager (Azure Biosystems 300Q, Dublin, CA, USA).

### 4.11. Statistical Analysis

Statistical analysis was performed using GraphPad Prism (Graphpad, San Diego, CA, USA). All results are expressed as either mean ± standard deviation (SD) or mean ± standard error of the mean (SEM) with two-tailed *p* values < 0.05 being considered as statistically significant. Differences between the groups were compared using Fisher’s exact test or a non-parametric test by Mann–Whitney U or Kruskal–Wallis as appropriate.

## Figures and Tables

**Figure 1 ijms-24-05186-f001:**
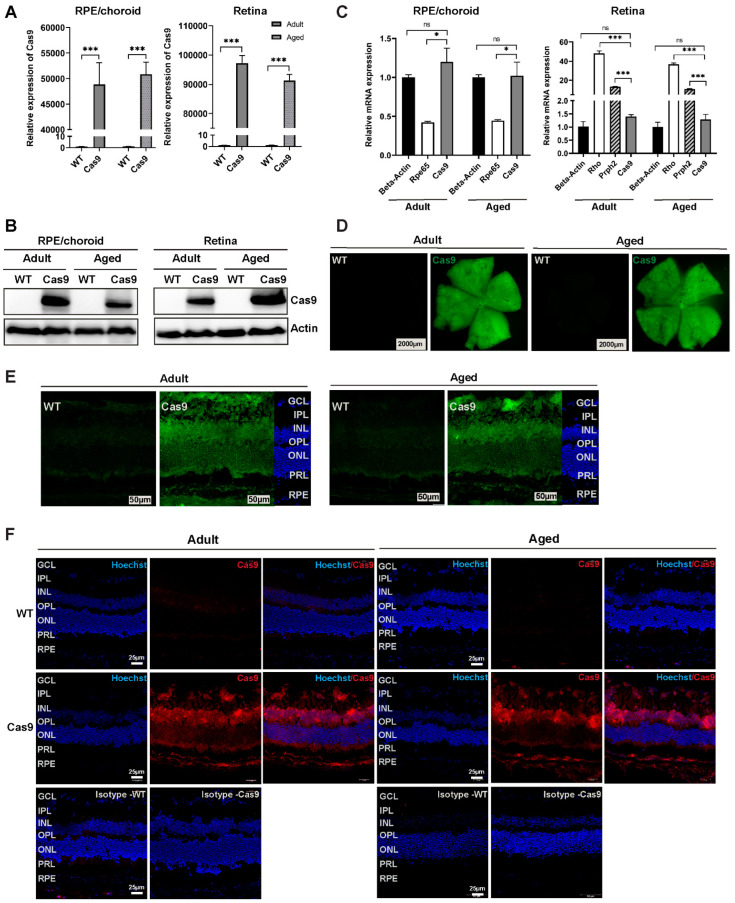
Evaluation of SpCas9 expression in RPE/choroid and retina of WT and Cas9 mice. (**A**) qPCR analysis of RPE/choroid and retinal tissues reveal abundant Cas9 expression in the adult and aged Cas9 mice (n = 5–6 per group; *** *p* < 0.001). Statistical significance was determined by Mann–Whitney U test; error bars depict SEM. (**B**) Representative Western blots show Cas9 expression in RPE/choroid and neural retina of adult and aged Cas9 mice. β-actin served as the loading control. (**C**) Relative expression of Cas9 in RPE/choroid and retina of adult and aged Cas9 mice with respect to the beta-actin, Rpe65 (RPE/choroid), and Rho; Prph2 (retina) was determined by qPCR (n = 5–6 per group, * *p* < 0.05, *** *p* < 0.001, ns, no significance). All qPCR data were normalized to GAPDH and error bars were represented as SEM. (**D**) Fluorescence images of retinal flat mounts from adult (left) and aged (right) mice showing Cas9-EGFP expression only in Cas9 mice (scale bars, 2000 μm). (**E**) Representative fluorescence images of retinal cross sections of adult (left) and aged (right) mice showing widespread Cas9 expression across different retinal layers. Representative WT mice retinal cross sections exhibit mild autofluorescence (scale bars, 50 μm). (**F**) Representative immunofluorescence images of the retina from WT and Cas9 mice showing ubiquitous Cas9 expression across retinal layers in adult and aged Cas9 mice. Adult and aged WT and isotype controls stained negative for anti-SpCas9 antibody. GCL, ganglion cell layer; IPL, inner plexiform layer; INL, inner nuclear layer; OPL, outer plexiform layer; ONL, outer nuclear layer; PRL, photoreceptor layer, RPE; retinal pigment epithelium. Hoechst staining (Hoechst 33342, blue) in both WT and Cas9 mice represent intact nuclei of retina. Scale bar: 25 μm.

**Figure 2 ijms-24-05186-f002:**
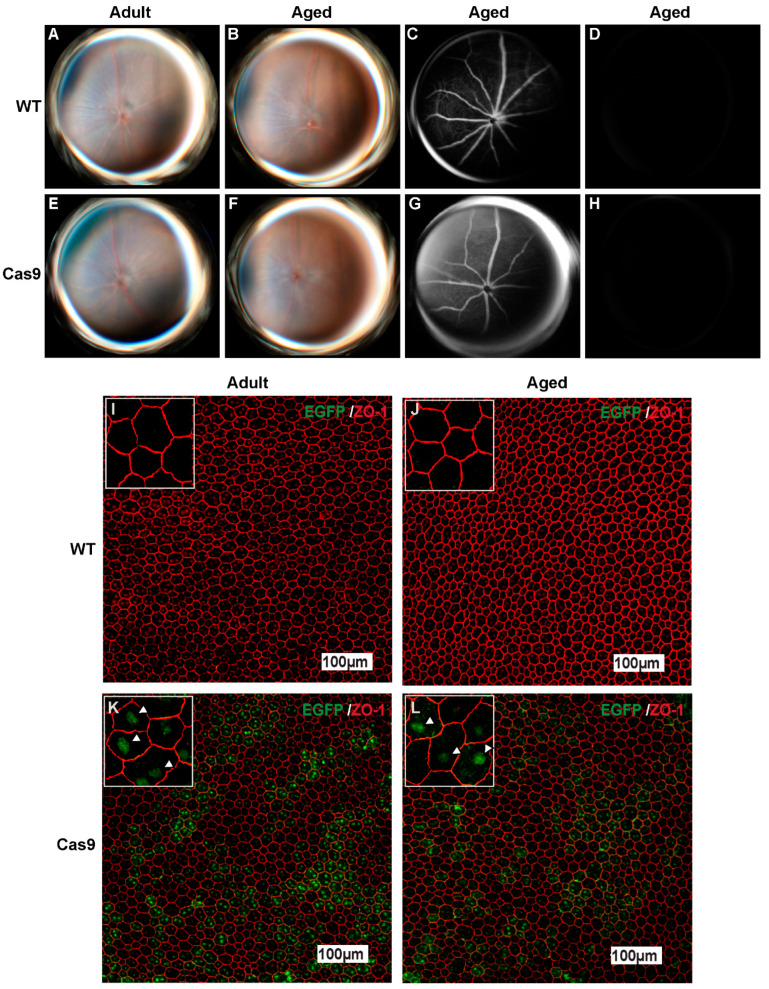
Phenotypic comparison between wildtype and Cas9 mouse eyes. (**A**,**E**) Representative fundus photographs of adult Cas9 mice (**E**) appear to be normal and healthy and comparable to those of the adult WT mice (**A**). (**B**,**F**) Representative fundus photographs of aged Cas9 mice (**F**) show normal fundus and are comparable to those of the similarly aged WT mice (**B**). (**C**,**G**) Fluorescein angiography of fundus showed no vascular defects in aged Cas9 and WT mice. (**D**,**H**) Fundus autofluorescence imaging showed absence of autofluorescence in both aged Cas9 and WT mice. (**I**,**J**) RPE morphology was visualized by immunostaining for ZO-1 (red) on *R*PE/choroid flat mounts of adult and aged WT mice. (**K**,**L**) RPE of adult Cas9 mice and age-matched controls showed typical hexagon pattern with ZO-1 staining (red) and nuclear-localized signal from EGFP (green).White arrowheads in images from the inset in K and L indicate nuclear-localized EGFP. Scale bar: 100 μm.

**Figure 3 ijms-24-05186-f003:**
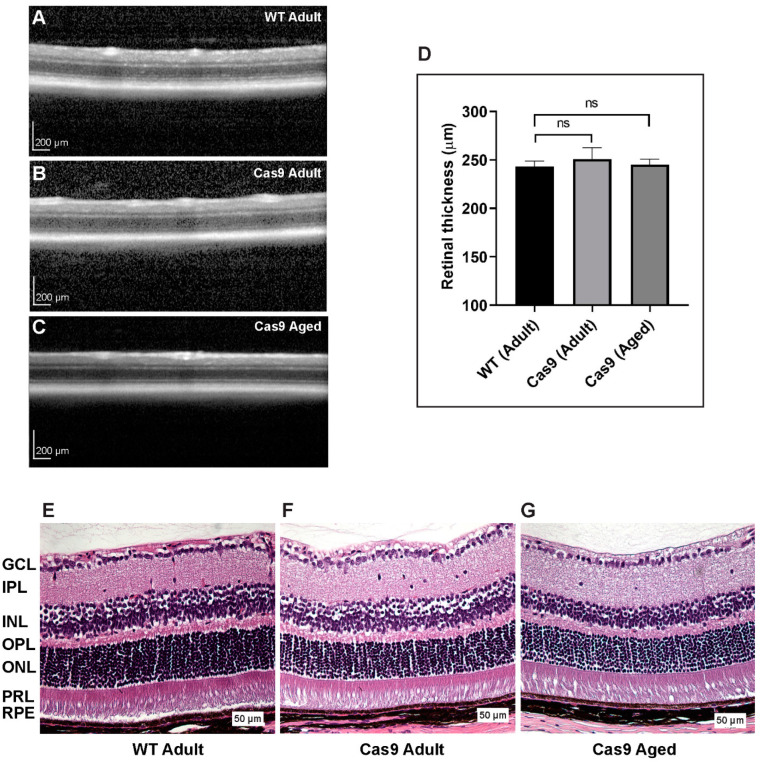
Comparison of WT and Cas9 mice retinal structure using SD-OCT imaging and histostaining. (**A**–**C**) Representative SD-OCT images from WT and Cas9 mice reveal normal retinal layers with no indication of retinal thinning or degeneration. (**D**) No statistically significant differences were observed in total retinal thickness between WT and Cas9 mice (*p* = 0.53, ns, no significance, Mann–Whitney test). (**E**–**G**) H&E staining showed all the major retinal layers and RPE/choroid to be intact in both WT and Cas9 mice. The retinal layers of the Cas9 mice were well aligned with the WT mice retinal layers represented by GCL, ganglion cell layer; IPL, inner plexiform layer; INL, inner nuclear layer; OPL, outer plexiform layer; ONL, outer nuclear layer; PRL, photoreceptor layer, RPE; retinal pigment epithelium. Scale bar: 50 μm.

**Figure 4 ijms-24-05186-f004:**
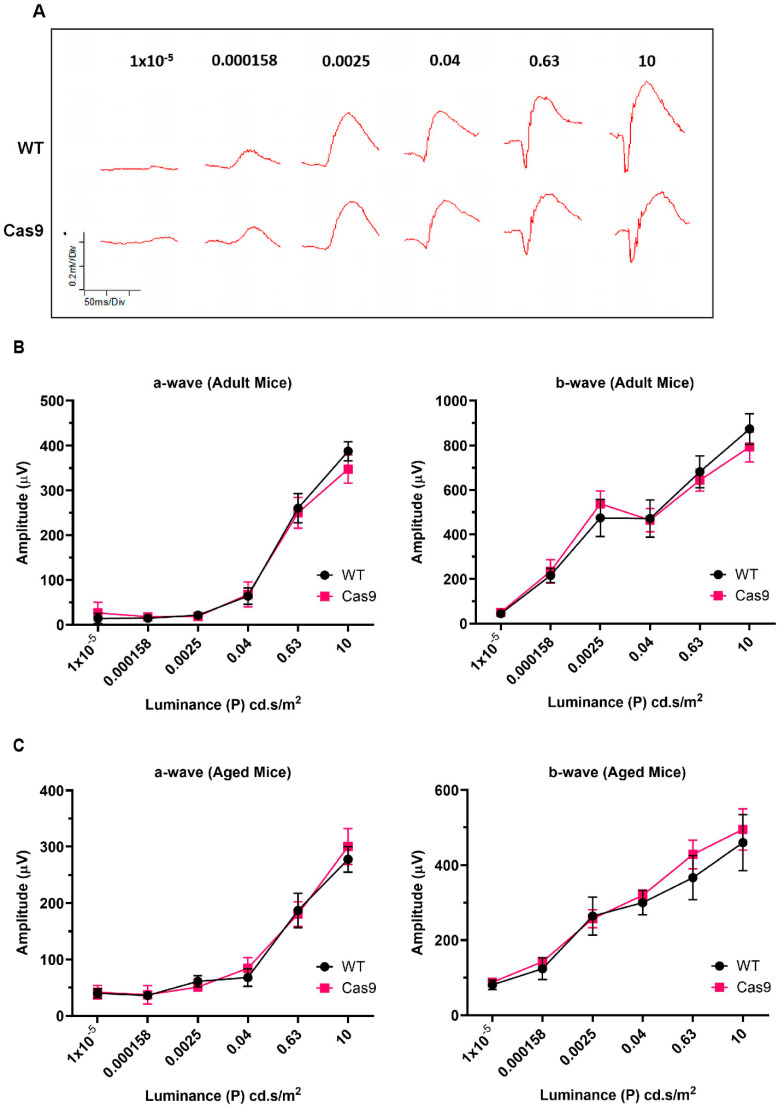
ERG analysis of retina function in Cas9 mice (**A**) Representative waveforms from ERGs of adult WT and Cas9 mice. (**B**,**C**) Scotopic a- and b-wave amplitudes, respectively, of adult and aged WT and Cas9 mice as a function of light intensity. Error bars represent SD (n = 6–8 per group). The a-wave and b-wave amplitudes at different flash intensities did not show any statistically significant (Mann–Whitney test; *p*-value < 0.01 considered significant) differences between age-matched WT (black) and Cas9 mice (red).

**Table 1 ijms-24-05186-t001:** *p*-values from statistical comparison of electroretinogram amplitudes of WT and Cas9 mice.

Flash Intensity ((P) cd.s/m^2^)	a-Wave (4–6 Months)	b-Wave (4–6 Months)	a-Wave (10–14 Months)	b-Wave (10–14 Months)
0.00001	0.4136	0.1812	0.6991	0.2403
0.000158	0.7546	0.5728	0.9372	0.132
0.0025	0.4908	0.1419	0.0649	>0.9999
0.04	0.662	0.7546	0.132	0.1797
0.63	0.5728	0.345	0.6991	0.0931
10	0.0513	0.0732	0.3095	0.2403

**Table 2 ijms-24-05186-t002:** The implicit times of a-wave and b-wave in adult and aged mice did not show significant difference between WT and Cas9 groups.

Flash Intensity ((P) cd.s/m^2^)	a-Wave Implicit Time (ms)		*p*_Value	b-Wave Implicit Time (ms)		*p*_Value	a-Wave Implicit Time (ms)		*p*_Value	b-Wave Implicit Time (ms)		*p*_Value
	Adult-WT	Adult-Cas9		Adult-WT	Adult-Cas9		Aged-WT	Aged-Cas9		Aged-WT	Aged-Cas9	
0.00001	43.33 ± 2.84	43.5 ± 3.44	0.8168	111.5 ± 1.59	115.75 ± 1.74	0.0846	29.52 ± 1.40	31.34 ± 0.70	0.513	90.26 ± 3.78	97.73 ± 1.56	0.2403
0.000158	26 ± 4.74	31 ± 3.66	0.5851	110.67 ± 2.22	114.63 ± 1.70	0.0949	35.27 ± 1.07	33.81 ± 0.87	0.3095	104.05 ± 2.52	102.71 ± 1.51	0.6991
0.0025	26.33 ± 0.33	24.88 ± 2.51	0.1951	111 ± 3.48	112.75 ± 1.97	0.9191	34.74 ± 2.74	33.65 ± 1.66	0.5087	98.94 ± 2.94	95.57 ± 2.34	0.1797
0.04	25.67 ± 0.76	26.88 ± 0.91	0.4659	79.83 ± 1.76	86.88 ± 4.09	0.3097	29.28 ± 1.45	26.86 ± 0.85	0.0584	85 ± 1.79	81.47 ± 1.58	0.132
0.63	20.17 ± 0.65	22 ± 0.78	0.0696	83.67 ± 0.88	87.75 ± 3.48	0.7529	23.07 ± 1.42	21.2 ± 0.43	0.2359	70.13 ± 4.51	72.6 ± 1.97	0.9372
10	10.17 ± 0.17	10.75 ± 0.25	0.1189	75.83 ± 1.25	81.88 ± 3.15	0.2384	18.72 ± 0.72	17.17 ± 0.61	0.145	70.33 ± 8.07	63.17 ± 2.23	0.5584

## Data Availability

Data are available within the article and Appendix A.

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
