# Peer review of "Long-Term Evaluation of Retinal Morphology and Function in Rosa26-Cas9 Knock-In Mice"

_ijms, 2023, doi:10.3390/ijms24065186_

Round 1

Reviewer 1 Report

The Rosa26-SpCas9.GFP knock-in mouse offers great potential to the gene editing field. Previous characterisation of this mouse model did not include ocular observations, which the authors present here. Understanding the ocular SpCas9 expression profile in this model is of interest and a lot of the data included are encouraging and indicate the mouse does express SpCas9 in the neural retina and RPE but that the mouse maintains a WT phenotype.

However, given that the novelty of this work lies in revealing the SpCas9 profile of expression in the different layers of the neural retina, having good quality retinal sections with different co-staining presentations would be appropriate. The current image quality indicates relatively poor retinal sections that do suggest SpCas9 in the GCL, INL and potentially the inner segments of photoreceptors but the image quality is not ideal. I would encourage the authors to review their imaging protocol to be sure the sections are not overexposed and also to co-stain with cell-specific markers (e.g. for GCL, INL and PR inner segments). Greater understanding of the cell expression profile in the neural retina would be highly valued, particularly as the majority of gene editing will involve targeting genes expressed in the photoreceptor cells therefore if the authors can clarify whether the SpCas9 expression is reliably detected (and sustained) in these cells in this mouse model, that would be important data to share. If this mouse model does not present much SpCas9 expression in photoreceptors, this is still an important finding and should be discussed. 

On a general note, I disagree that mice aged 4-6 months should be referred to as young, I think a cohort at 1-2 months of age would suit this categorisation better. Young/old/4-6 months/10-14 months are interchanged throughout the paper. Perhaps define as “adult” and “aged” and be consistent thereafter?

When looking at SpCas9 mRNA levels, comparison to WT eyes would be expected to show a dramatic difference but how do the SpCas9 levels compare to other retinal genes? A small panel of 3 or 4 selected retinal genes could be used for this.

Figure 1 should have data for both age groups of mice, it is important to know if Cas9 expression is maintained. For Figure 1E, a secondary-only image for the Cas9 mouse section would be helpful. As mentioned above, the retinal tissue appears to have poor structure and the red channel seems over-exposed. Also in Figure 2J & L, the red channel seems over-exposed, this needs to be reviewed and perhaps adjusted. Where is the GFP expression and SpCas9 co-staining in the RPE flat mounts? Line 279 indicates EGFP is shown on the flat mounts but this does not seem to be the case.

For Figure 3, I’m not clear where the OCT measurements were taken and how many were taken per eye. Were they radial scans with a single measurement taken for each eye on the vertical axis (superior retina?) or were multiple measurements taken at radial positions and averaged? Also, WT vs Cas9 statistical comparisons of thickness measurements should be made between age-matched cohorts. Caliper positions on an example image would be of interest.

Were the ERG  implicit times comparable for WT and Cas9 mice? 

Specific edits to the text/structure:

  • be consistent with knock-in (not knockin)
  • line 73/74 - “This Cas9-knockin mice” - perhaps change to “The Cas9 knock-in mouse model…”
  • loss of sentence structure between lines 97 & 98 (Figure 1 needs to be moved)
  • line 122 - states rd1 and rd8 mutations were screened for but the methods only describe rd8
  • line 298 - I don’t think the data presented demonstrate SpCas9 expression is safe in the RPE/choroid and retina
  • line 369 - it won’t be obvious to a lot of people how RPE/choroid flat mounts are prepared so more methodological detail might be required

Reviewer 2 Report

This very well written article presents extensive data on yound and old Rosa26-Cas9 ki mice that will be a valuable resource for preclinical in vivo gene editing sturdies. The results are nicely presented and discussed.

Could the authors please state clearly in the last paragraph of the introduction:

1. What is the difference between the Rosa-Cas9 ki mouse and the SpCas9 mouse described in Ref 24, Platt et al., Cell, 2014.

2. Brief description how the Rosa-Cas9 ki mouse line was generated and in which studies these mice have already been used 'outside' the eye.

There is also a inconsistency between Results and Methods: in Methods genotyping for the rd8 mutation is described, but not for the rd1. However, in the Results section rd1 genotyping is also mentionned. From the histology it is obvious that the rd1 mutation is not present but please clarify whether rd1 genotyping has been done. 
